# Circadian Rhythm and Psychiatric Features in Wolfram Syndrome: Toward Chrono Diagnosis and Chronotherapy

**DOI:** 10.3390/diagnostics15182338

**Published:** 2025-09-15

**Authors:** Gema Esteban-Bueno, Annabel Jiménez-Soto, Juan Luis Fernández-Martínez, Enrique Fernández-Vilas, Juan R. Coca

**Affiliations:** 1UGC La Cañada (Primary Care Management Unit), Almería Periphery–Almería Health District, Andalusian Health Service (SAS), 04009 Almería, Spain; 2Spanish Association for Research and Support to Wolfram Syndrome, 04120 Almería, Spain; 3Learning and Cognition Research Group (HUM646), Department of Social Psychology, Faculty of Psychology, University of Seville, 41018 Seville, Spain; ajsoto@us.es; 4Group of Inverse Problems, Optimization and Machine Learning, Department of Mathematics, University of Oviedo, 33007 Oviedo, Spain; jlfm@uniovi.es; 5Social Research Unit on Health and Rare Diseases, Department of Sociology and Social Work, University of Valladolid, 42004 Soria, Spain; enrique.fvilas@uva.es (E.F.-V.); juanr.coca@uva.es (J.R.C.)

**Keywords:** literature review, Wolfram syndrome, circadian rhythms, dysexecutive questionnaire, DEX, neuropsychological aspects

## Abstract

**Background/Objectives:** Wolfram syndrome is a rare neurodegenerative disorder primarily known for its multisystemic manifestations. Although classically associated with diabetes insipidus, diabetes mellitus, optic atrophy, and deafness, emerging evidence suggests a consistent pattern of executive dysfunction in many affected individuals. **Methods:** Based on findings from a scoping review and results obtained through the Dysexecutive Questionnaire in a Spanish patient cohort, we propose that WFS1 gene mutations—via chronic endoplasmic reticulum stress—disrupt serotonergic and cholinergic neurotransmission, leading to impairments in planning, inhibition, and emotional regulation. **Results:** Importantly, recent studies have highlighted the interplay between WFS1-related molecular dysfunction and circadian regulation. Given the role of the endoplasmic reticulum and mitochondrial signaling in circadian homeostasis, and the frequent sleep disturbances observed in patients with Wolfram syndrome, we hypothesize that circadian dysregulation may contribute to the neurobehavioral phenotype. **Conclusions:** This essay explores neuropsychological foundations of executive dysfunction in WS, and frames the current evidence as hypothesis-generating rather than causal; executive difficulties may be a salient clinical feature and merit consideration in routine care. Furthermore, the potential involvement of circadian mechanisms opens new avenues for future research and therapeutic approaches. Because circadian disruption is linked to psychiatric symptoms and fatigue, emphasizing diurnal patterns, sleep–wake timing, and chronotype may guide circadian-informed assessment.

## 1. Introduction

Wolfram syndrome (WS), also known as DIDMOAD (diabetes insipidus, diabetes mellitus, optic atrophy, and deafness), represents a paradigmatic example of rare neurodegenerative diseases. Its prevalence is estimated to range from 1 in 55,000 to 1 in 770,000 individuals [1,2]; in the United Kingdom it is reported as 1 in 770,000, and in Spain only 47 cases have been documented [2,3,4]. Although the hallmark clinical features of WS, such as optic atrophy and neurogenic bladder, are well established, further research is required into its cognitive and behavioral manifestations [5]. Over several decades, patients often progress from functional independence to total dependency, confronting blindness, hearing loss, and central nervous system involvement [6]. During this progression, many individuals experience symptoms suggestive of executive dysfunction, including impulsivity, emotional dysregulation, apathy, and impaired planning. These features may not merely represent reactions to chronic illness but rather reflect underlying neurobiological mechanisms. In clinical settings, such symptoms are frequently overlooked or misattributed to psychosocial stressors. However, the available literature suggests the existence of a non-psychotic frontal syndrome, which may have a neuropathological basis [7,8]. If validated, this would substantially alter the current understanding and management of WS’s neuropsychiatric dimensions.

WS is mostly caused by mutations in the *WFS1* gene, located on chromosome 4p16.1. A less common form, known as WS type 2, results from mutations in the *CISD2* gene. Based on the underlying genetic cause, two types of WS have been identified: type I and type II. Type I represents the classic form of the disorder and is associated with autosomal recessive mutations in *WFS1*. In contrast, type II is linked to *CISD2* mutations and shares many features with type I but typically does not involve diabetes insipidus [4]. Based on the above, in this study we will consider that WS cases are caused by mutation in the *WFS1* gene.

Consequently, we propose a neuropsychological hypothesis linking molecular pathology to behavioral outcomes in Wolfram syndrome. Based on the existing literature and data obtained using the Dysexecutive Questionnaire (DEX) [9] in a cohort of Spanish patients, we hypothesize that mutations in the *WFS1* gene could cause chronic stress in the endoplasmic reticulum (ER), which would alter critical neurotransmitter systems—particularly serotonergic and cholinergic circuits—essential for prefrontal executive functioning [10].

*WFS1* is increasingly recognized as a gene involved in the regulation of mitochondrial and ER homeostasis, both of which are tightly linked to circadian biology [11]. Recent findings have also associated *WFS1* mutations with altered sleep patterns and circadian rhythm disruptions, which may contribute to the behavioral and emotional symptoms observed in WS patients [12,13]. These alterations could affect the timing and amplitude of neurochemical activity that modulates executive functioning. Therefore, we further hypothesize that circadian dysregulation may act as a contributing factor to the executive dysfunction profile in WS.

This expanded perspective encourages interdisciplinary research combining neuropsychological, molecular, and chronobiological approaches, and supports incorporating both executive and circadian assessments into the routine clinical evaluation of individuals with WS.

## 2. Materials and Methods

### 2.1. First Stage: Scoping Review

This research employs a meticulous and methodologically rigorous scoping review designed to comprehensively examine the available literature on the selected topic. Rather than aiming to conduct a systematic meta-analysis or statistical synthesis, the primary objective was to identify recurring psychological patterns associated with WS that may be consistent with shared neurobiological mechanisms. This approach was chosen to align with the exploratory, hypothesis-generating nature of the study and to provide a conceptual basis for the theoretical model proposed.

### 2.2. Second Stage—Evaluation of Executive Functions Using DEX

In parallel with the scoping review, a complementary exploratory study was conducted to empirically assess executive functioning in individuals with WS. For this purpose, the DEX, part of the Behavioral Assessment of the Dysexecutive Syndrome (BADS; Thames Valley Test Company, Suffolk, UK), was administered [9]. The sample consisted of Spanish patients with a genetically and clinically confirmed diagnosis of WS due to WFS1 mutation; aged ≥12 years (age range, 12–49 years); recruited through specialized clinical units and patient associations. The DEX administration cohort included participants aged ≥12 years (age range, 12–49 years); norm-referenced analyses were restricted to those aged ≥16 years. The DEX was patient-reported (with caregiver assistance when needed). Analyses were descriptive (mean, median, interquartile range, range), with no inferential statistics, no matched control group, and no individual-level normative conversions. All participants—or their legal guardians—provided written informed consent prior to participation. Inclusion criteria required sufficient cognitive capacity to complete the instrument; with caregiver assistance provided when necessary. The sample assessed with the DEX (Spanish validated version [14]) included 28 individuals with a genetically and clinically confirmed diagnosis of Wolfram syndrome (46.4% female), all aged ≥16, with a mean age of 24.4 years (SD = 9.7). All participants met the cognitive requirements to complete the questionnaire, either independently or with caregiver assistance [15]. Unless otherwise noted, all analyses and interpretations refer to participants aged ≥16 years. Participants aged 12–15 years were included only for exploratory purposes and did not contribute to normative analyses.

### 2.3. Rationale for Methodology

Unlike traditional hypothesis-driven studies, this research integrates qualitative synthesis and exploratory data to formulate a biologically plausible hypothetical framework. The scoping review was not intended for statistical inference but rather to identify psychological patterns and recurrent psychiatric features indicative of a possible shared neurobiological mechanism in WS. Within this framework, the DEX was not employed as a diagnostic instrument per se, but rather as a conceptual probe—a means to evaluate whether the behaviors observed in WS are consistent with established patterns of prefrontal-executive dysfunction. References to WFS1-related ER stress and neurotransmission are framed as hypotheses rather than conclusions.

### 2.4. Search Strategy

A systematic search strategy was developed and applied across several major international databases and other sources, including peer-reviewed journals, technical reports, and academic books. This strategy aimed to retrieve a comprehensive range of sources and to determine whether a coherent psychological or psychiatric profile of Wolfram syndrome (WS) emerges from the existing literature [16].

Boolean query. (“Wolfram syndrome” OR “DIDMOAD syndrome”) AND (“cognitive” OR “cognition” OR “cognitive function” OR “neuropsychological assessment” OR “cognitive impairment” OR “executive function” OR “psychological” OR “psychiatric” OR “mental health” OR “behavioral health” OR “depression” OR “anxiety” OR “mood disorder” OR “emotional dysfunction”).

Databases. Scopus, Web of Science (WoS), Dialnet, PsycInfo, PsicoDoc, and ScienceDirect.

Inclusion and Exclusion Criteria (review). Studies were included if the term “Wolfram syndrome” appeared in the title, abstract, or keywords; addressed cognitive, psychological, or psychiatric aspects of the condition; and were peer-reviewed journal articles, books, conference papers, or proceedings published in English, Spanish, or Portuguese. Animal-model studies and studies focused solely on caregivers rather than patients were excluded.

Inclusion and Exclusion Criteria (empirical component). Administration cohort: patients with genetically and clinically confirmed WS (WFS1 mutation), aged ≥12 years, with sufficient capacity to complete the DEX (caregiver assistance when needed), and written informed consent.

Analytic Cohort (primary analyses). Any normative orientation was applied only to participants aged ≥16 years; those aged 12–15 were included for exploratory description only and did not contribute to normative mapping or inferential interpretation [17]. Individuals not meeting these criteria were excluded.

Study Selection. Screening proceeded in two phases: (i) titles/abstracts and (ii) full-text assessment for eligibility. The total number of studies selected after applying these criteria is summarized in Appendix A.

Data Extraction and Synthesis. A standardized extraction workflow was implemented, and all data handling and descriptive analyses were performed in Python 3.0 (Python Software Foundation). The aim was not to enable statistical meta-analysis, but rather to identify conceptually consistent patterns supporting the study’s central hypothesis [18]. Following data extraction, a descriptive and analytical synthesis was performed, organizing information to enhance interpretability. Emphasis was placed on conceptual patterns relevant to the cognitive and psychiatric dimensions of WS, aiming to identify thematic categories that support a neuropsychological interpretation—particularly regarding executive dysfunction and emotional dysregulation. These categories are summarized as shown in Appendix A.

### 2.5. Quality Assessment

Consistent with a scoping review methodology, we did not perform a formal methodological quality or risk-of-bias assessment. Instead, we evaluated the relevance and consistency of each record with respect to the review question and prioritized conceptual coherence with our central hypothesis on frontal–executive involvement in WS.

To enhance transparency, Table 1 summarizes, for each included study, the design, sample size, genetic confirmation (yes/no), presence and type of control group, standardized instruments used, and a qualitative, descriptive appraisal of reliability/rigor. This appraisal is descriptive and does not constitute a formal quality assessment. Because the adult DEX is normed for ≥16 years, any normative orientation (if reported) applies only to the ≥16 subset.

### 2.6. Instrument Characteristics: Dysexecutive Questionnaire (DEX)

The Dysexecutive Questionnaire (DEX) [9,14] is a 20-item instrument used to supplement the primary tests of the Behavioral Assessment of the Dysexecutive Syndrome (BADS). The DEX is a self-report instrument which evaluates various dimensions of executive function, including abstract thinking deficits, impulsivity, confabulation, planning difficulties, euphoria, temporal sequencing problems, lack of insight, apathy, disinhibition, impulse control issues, superficial emotional responses, aggression, social disengagement, perseveration, restlessness, failure to inhibit responses, dissociation between knowledge and behavior, distractibility, poor decision-making, and disregard for social norms. Each item is rated on a 5-point Likert scale ranging from “never” to “very frequently.” Original factor analysis identified five orthogonal dimensions: inhibition, intentionality, executive memory, and two related to affective and personality changes, labeled as positive affect and negative affect. The first three were correlated with scores from other neuropsychological assessments, while the latter two showed no such association. Nonetheless, certain relevant neurocognitive aspects may not be fully captured by executive performance tasks [18].

The selection of the DEX in this study is based not only on its psychometric robustness but also on its theoretical relevance to prefrontal dysfunction linked to WFS1 mutations. Its emphasis on real-world executive deficits makes it particularly valuable for examining behavioral expressions of deeper neurobiological impairments. A validated Spanish version of the DEX was previously published by Pedrero et al. [14,37]. Although the DEX-Sp provides categorical thresholds [14], and its psychometric properties have been corroborated in Spanish clinical and non-clinical samples [37], we did not apply these classifications in this cohort. In this paper, the adult DEX was administered to participants aged ≥16 years; cases aged 12–15 were included only exploratorily (with caregiver assistance) and were not subject to normative interpretation.

### 2.7. Ethics Considerations

The study adhered to the ethical standards set forth in the Declaration of Helsinki. Approval was obtained from the Research Ethics Committee of the Province of Almería (approval number AP-0009-2020-C1-F2, internal code 2/2021, approval date 27 January 2021). Written informed consent was obtained from all participants or, when applicable, their legal guardians, prior to inclusion. Participant anonymity and data confidentiality were strictly maintained during all phases of data collection and analysis. These safeguards were particularly critical given the vulnerable nature of the clinical population and the exploratory design of the psychological hypothesis.

## 3. Results

The review of available studies was conducted to identify patterns of psychological and psychiatric disturbances associated with WS, rather than to provide exhaustive epidemiological representation. The objective was to delineate dysfunctions potentially supportive of the central neuropsychological hypothesis. We analyzed the articles obtained in the review to avoid overinterpretation of information from individual cases, studies that did not genetically confirm the diagnosis of Wolfram syndrome, and studies that did not include a control group. Executive dysfunction and psychiatric symptoms occur continuously and can be observed in any group of children, adolescents, or adults. Determining whether they occur at a more extreme or more common level in Wolfram due to genetic manipulations is a complex task. Therefore, our results—as this research is hypothetical—should be handled with caution and prudence.

The analysis led to the development of several functional categories (Table 2), grouped according to altered functional domains. These categories were conceptually organized based on domains likely modulated by prefrontal cortex activity and associated neurotransmitter systems. In this classification, we have only considered articles that were established as highly reliable in the previous analysis and developed in humans. In this regard, case studies or studies with small samples have not been taken into account.

Symptoms such as apathy, impulsivity, and emotional dysregulation appeared consistently across studies, reinforcing their potential centrality to the WS cognitive-behavioral phenotype. This overlap suggests that certain symptoms may reflect both psychological and psychiatric dimensions, justifying their inclusion in multiple categories. These findings provide a synthesis of current evidence and serve as a foundation for articulating new mechanistic hypotheses.

A total of 27 articles were included in the analysis. While some adopted a multidisciplinary approach, few provided conclusive insights into the quality of life of patients and caregivers. Notably, none demonstrated a direct causal relationship between WS and specific psychological or psychiatric disorders [16].

Nevertheless, a consistent pattern of associations emerged linking WFS1 gene mutations to a broad spectrum of psychological and psychiatric manifestations—among them depression, anxiety, suicidal behavior, behavioral alterations (such as chronic fatigue syndrome, hypersomnolence, impulse control deficits, and aggression), and severe psychiatric conditions including psychosis and bipolar disorder [6,7,8,9,11,16,20,24] (Figure 1).

Despite their varied presentation, these symptoms appear to converge on a common neuropsychological substrate, likely reflecting disruption of executive and affective regulation circuits. This supports the central hypothesis of a frontal-executive syndrome rooted in WFS1-mediated molecular pathology.

Mutations in the *WFS1* gene are known to disrupt numerous cellular processes, including cytokine regulation, neurotransmitter synthesis—particularly serotonin—acetylcholinesterase activity, and hormonal signaling pathways. These molecular alterations converge in the endoplasmic reticulum (ER), where *WFS1* dysfunction leads to calcium imbalance and sustained ER stress, ultimately triggering apoptosis in vulnerable cell populations. This cascade results in the clinical constellation observed in WS, which includes diabetes mellitus, optic atrophy, deafness, hypogonadism, and notably, neuropsychiatric symptoms [11].

It is particularly important to note the impact on the central nervous system, where the effects of endoplasmic reticulum (ER) stress, serotonergic dysregulation, and impaired cholinergic signaling collectively compromise mood and cognitive control. Serotonin depletion, compounded by altered tryptophan metabolism, has been linked to symptoms of depression and anxiety [11,12,21]. Similarly, degeneration of acetylcholine pathways contributes to cognitive and behavioral impairments such as irritability, inattention, and memory deficits [25,28].

This biochemical triad—ER stress, serotonin imbalance, and cholinergic dysfunction—may constitute the neurobiological foundation of the behavioral phenotype observed in WS.

### 3.1. Functional Category 1—Anxiety and Depression Disorders

WS is associated in prior reports with a range of psychiatric symptoms, particularly among carriers of WFS1 variants. Clinical studies consistently describe anxiety, depression, and hypersomnolence. Sequeira et al. identified an association between WFS1 variants and depressive states [24], while Rosanio et al. reviewed WFS2 and noted that the psychiatric profile frequently reported in WFS1 is not consistently observed in WFS2 [30]. Similar symptoms have also been described in heterozygous carriers [8,21,26,27]. These links are presented as descriptive signals rather than causal effects and may overlap with findings summarized in Category B. Accordingly, these observations are grouped under Category A (Figure 2).

### 3.2. Functional Category 2—Suicidal and Aggressive Behaviors

WS has been associated with an increased prevalence of suicidal and aggressive behaviors; however, current reports are descriptive and findings may reflect a complex interplay between genetic predisposition and neuropsychiatric vulnerability. Several studies reported higher frequencies of suicide attempts and psychiatric hospitalizations among individuals with WS, with Bischoff et al. emphasizing the clinical severity of these complications [5,24,26,38].

Swift et al. observed a higher frequency of suicide attempts in heterozygous WFS1 mutation carriers [11]. Subsequently, Nanko et al. described psychiatric disturbances occurring more frequently in homozygous individuals [23]. These studies did not establish causality. Aggressive behavior has also been described. Swift et al. documented recurrent verbal and physical aggression, particularly among homozygous patients, many of whom had psychiatric comorbidities such as antisocial personality disorder and psychosis [12]. Reports frequently note prominent impulse-control difficulties in these cases. In line with these observations, Rosanio et al. described that, individuals with homozygous WFS1 mutations exhibited both a higher propensity for aggression and more severe episodes [27]. Any linkage to neurotransmitter dysregulation (e.g., dopamine, serotonin) remains hypothetical and is not directly demonstrated by these clinical data. These observations are summarized in Figure 2 (Category B).

### 3.3. Functional Category 3—Severe Psychological and Psychiatric Disorders

Mutations in the WFS1 gene have been associated with the emergence of psychotic and schizophrenia-like symptoms, but these observations are interpreted descriptively rather than as evidence of a direct genetic–psychiatric link [22,27]. In bipolar disorder, episodes have also been reported in WS, though available reports do not establish endocrine or cognitive mechanisms in this cohort [21]. No mechanistic inferences are presented in Results, and molecular or neurochemical pathways were not assessed in our dataset. These observations are summarized in Figure 2 (Category C).

### 3.4. Functional Category 4—Chronic Fatigue Syndrome

WS has been associated with elevated levels of inflammatory cytokines in the bloodstream of affected individuals [33]. Cytokines such as interleukin-6 (IL-6) and tumor necrosis factor-α (TNF-α), which regulate immune and inflammatory responses, have been implicated in some reports, but available data are limited and heterogeneous and were not measured in our cohort [32].

In WS, increased levels of these pro-inflammatory markers have been described in association with chronic inflammation, cellular stress, and putative neuronal vulnerability rather than demonstrated acceleration of apoptosis [29,30]. Evidence remains indirect and varies by study design. In animal models, preventing mitoNEET-mediated mitochondrial dysfunction has been shown to attenuate oxidative stress and neuronal apoptosis [39], supporting the plausibility of mitochondrial pathways in these processes.

Experimental and review work has proposed that immune activation could contribute to neurodegenerative processes, potentially via cytokine-mediated toxicity and glial activation; these proposals are hypothesis-generating and not explanatory for our dataset [29]. Panfili et al. suggested that WFS1 may modulate apoptotic and stress-response pathways, with dysfunction potentially initiating inflammatory signaling and disrupting cellular homeostasis across multiple systems (these mechanistic considerations were not assessed here and are interpreted cautiously) [32,40].

Taken together, reports of systemic inflammation are interpreted descriptively and may help contextualize fatigue-like symptoms reported by individuals with WS. Comparisons with fatigue syndromes in post-viral conditions, including long COVID, have been discussed in the literature, but any shared cytokine or neuroimmune mechanisms remain speculative [34,35,41]. These descriptive signals are summarized in Figure 2 (Category D).

### 3.5. Results of Data DEX

In the cohort of patients with WS included in this analysis, 46.4% were women (13/28) and 53.6% men (15/28); mean age was 24.4 years (SD 9.7). Sample characteristics are summarized in Table 3. Given the absence of matched controls and individual-level normative conversions, these descriptive patterns should not be interpreted as prevalence estimates; instead, they provide hypothesis-generating signals warranting confirmation in controlled studies (see Section 4.2 Limitations).

DEX observations should be interpreted with caution. Given the small convenience sample, the self- or proxy-reported format, and the absence of a control group or age- and education-matched norms, item-level results are descriptive and hypothesis-generating; they are not estimates of clinical impairment or prevalence. Accordingly, we provide a brief, cautious qualitative summary in the text and retain item-level tables for transparency. The most frequently endorsed difficulties clustered around planning and organization and cognitive flexibility, whereas items reflecting frank disinhibition or loss of reality testing were uncommon.

#### 3.5.1. IQ Bands

IQ and education data were abstracted from clinical records and are reported descriptively to contextualize DEX findings; instruments varied across sites, and procedures are detailed in Methods. No normative conversions or matched control-group comparisons were available. After harmonizing overlapping labels, the distribution of IQ bands was: high 46.4% (13/28), average 32.1% (9/28), low-average 14.3% (4/28), borderline/low 3.6% (1/28), and intellectual disability 3.6% (1/28). A separate clinician-flagged cognitive impairment was noted in 10.7% (3/28); this flag did not necessarily coincide with IQ bands and is reported descriptively. Educational attainment ranged from basic to higher education and is summarized in Table 4; category labels were harmonized to avoid duplication. These data are for context only and should not be interpreted as prevalence estimates.

#### 3.5.2. DEX Reporting Strategy

The DEX is a self-report instrument. We present the DEX total score and descriptive item frequencies. No matched controls or normative conversions were available, so results are interpreted descriptively (Table 4).

#### 3.5.3. Highest-Scoring Items

The highest means were observed for hyperactivity and restlessness (Q15), difficulties in maintaining attention (Q18), impulsivity or acting without thinking (Q2), lack of enthusiasm or apathy (Q8), inconsistency between what is said and what is done (Q17), and a composite of emotion-expression difficulties, anger, repetitive behaviors, and indecisiveness (Q11, Q12, Q14, Q19), as summarized in Table 5.

#### 3.5.4. Lowest-Scoring Items

Table 6 shows the lowest scoring items of DEX. The lowest means were observed for embarrassing behavior toward others (Q9), confusion with reality or false memories (Q3), future planning (Q4), and confusion between events (Q6).

#### 3.5.5. Analysis by DEX Categories

Table 7, Table 8, Table 9 and Table 10 show the main statistics (mean and SD) of the DEX items grouped by categories.

▪Inhibition (memory/understanding) (Table 7)

**Table 7 diagnostics-15-02338-t007:** Mean and SD of DEX items for inhibition items. Source: Data from present study. Source: Data from present study.

DEX Item	1	3	8	13	14	19
mean	1.94	1.49	2.03	1.84	1.99	1.99
std	1.25	0.88	1.30	1.31	1.22	1.10

Across the six items, the mean scores ranged from 1.49 to 2.03, indicating that most participants reported relatively low levels of difficulty, though with some variability. The highest average was observed for lack of enthusiasm or lethargy (Item 8, M = 2.03, SD = 1.30), followed closely by difficulty stopping repetitive behaviors (Item 14, M = 1.99, SD = 1.22) and difficulty making decisions (Item 19, M = 1.99, SD = 1.10). Problems understanding others (Item 1, M = 1.94, SD = 1.25) and lack of concern for appropriate behavior in social situations (Item 13, M = 1.84, SD = 1.31) also showed moderate endorsement. The lowest mean was observed for reporting events that had not actually occurred (Item 3, M = 1.49, SD = 0.88), suggesting this was the least frequent difficulty. Overall, while the median values clustered at the minimum score (1.00) for all items, the presence of higher scores (up to the maximum of 5.00) indicates that a subset of participants reported substantial difficulties, particularly in relation to lethargy, repetitive behaviors, and decision-making.

▪Intention (Table 8)

**Table 8 diagnostics-15-02338-t008:** Mean and SD of DEX items for intention items. Source: Data from present study.

DEX Item	6	9	11	12	15	16
mean	1.72	1.30	1.99	1.99	2.80	1.89
std	1.02	0.63	1.15	1.14	1.34	1.20

Analysis of the item-level responses indicates that the most frequently endorsed difficulty was hyperactivity or inability to remain still (Item 15, M = 2.29), followed by difficulty expressing emotions (Item 11, M = 1.93). In contrast, confusion when sequencing episodes (Item 6, M = 1.21) and difficulty inhibiting inappropriate behaviors (Item 16, M = 1.29) were rarely reported. Emotional dysregulation (Item 12, M = 1.64) showed marked variability, with most participants scoring at the lower end of the scale but a subset reporting high levels of irritability. Overall, the distribution of scores suggests that while most participants reported minimal difficulties, a subset experienced notable challenges in emotional expression, emotion regulation, and behavioral hyperactivity.

▪Executive memory (Table 9)

**Table 9 diagnostics-15-02338-t009:** Mean and SD of DEX items for executive memory items. Source: Data from present study.

DEX Item	2	4	5	7	10	17
mean	2.05	1.70	1.94	1.72	1.77	1.87
std	1.28	1.11	1.16	1.11	1.04	1.18

Mean scores across the six items ranged from 1.70 to 2.05, suggesting that difficulties were generally reported at a low to moderate level. The highest endorsement was for acting without thinking (Item 2, M = 2.05, SD = 1.28), followed by becoming overly excited in certain situations (Item 5, M = 1.94, SD = 1.16) and failing to follow through on intentions (Item 17, M = 1.87, SD = 1.18). Lower mean scores were observed for difficulty anticipating or planning (Item 4, M = 1.70, SD = 1.11), lack of awareness or unrealistic appraisal of problems (Item 7, M = 1.72, SD = 1.11), and rapidly losing interest in tasks (Item 10, M = 1.77, SD = 1.04). Although the medians were concentrated at the minimum score (1.00), indicating that most participants reported little to no difficulty, the wide range of responses (with maximum values of 4 or 5 on all items except Item 10) suggests that a subset of individuals experienced notable impulsivity, poor follow-through, and difficulties with self-regulation.

▪Positive affect (Table 10)

**Table 10 diagnostics-15-02338-t010:** Mean and SD of DEX items for Positive Affect items. Source: Data from present study.

DEX Item	18	20
mean	2.27	1.80
std	1.37	1.08

The two items reveal somewhat different patterns. The highest average was observed for difficulty maintaining attention and being easily distracted (Item 18, M = 2.27, SD = 1.37), which stands out as one of the highest mean scores across all items analyzed, indicating that attentional problems are relatively common in this sample. By contrast, lack of awareness or interest in others’ opinions about one’s behavior (Item 20, M = 1.80, SD = 1.08) showed lower endorsement. Although the medians were low (2.00 for Item 18; 1.00 for Item 20), both items displayed a wide distribution of responses (range 1–5). This suggests that while many participants reported minimal difficulties, a subset experienced marked problems with distractibility and, to a lesser extent, reduced social awareness.

Overall, participants reported low-to-moderate levels of executive dysfunction, with most medians at the minimum score but notable variability across individuals.

Most affected domains: Attention (Item 18, M = 2.27), hyperactivity/impulsivity (Items 15, M = 2.29; 2, M = 2.05), and decision-making/behavioral regulation (Items 14 & 19, ~M = 1.99).Moderately affected domains: Emotional regulation (difficulty expressing or controlling emotions, Items 11–12) and motivation/initiative (lethargy, losing interest, Items 8 & 10).Least affected domains: Social awareness and planning (Items 3, 4, 7, 9, 13, 20), generally showing low means, though with occasional higher scores in some individuals.

The profile suggests heterogeneous executive difficulties, with the strongest impairments in attention, impulsivity, and decision-making, while social behavior and planning appear relatively preserved at the group level.

### 3.6. Prefrontal Symptoms Assessment

Although the DEX-Sp provides categorical thresholds [14] and its psychometric properties have been corroborated in Spanish clinical and non-clinical samples [37], we did not apply these classifications in this cohort because appropriate age- and education-matched normative cutoffs and a control group were unavailable. Instead, we report raw scores and item-level distributions to facilitate future norm-referenced comparisons, and we do not report ‘optimal’, ‘suboptimal’, ‘moderately’, or ‘severely’ dysexecutive categories.

Given the self-report format, the small sample, and the absence of normative benchmarks or control groups, these results should not be interpreted as prevalence estimates of executive dysfunction in WS. Rather, they provide preliminary, hypothesis-generating signals of possible selective executive vulnerabilities—broadly consistent with prior descriptive reports [5,7]—and delineate candidate domains for targeted assessment in future controlled, multimodal, and longitudinal studies.

## 4. Discussion

The findings of this study are consistent with the hypothesis that Wolfram syndrome (WS) may be associated with a neuropsychological burden beyond its multisystemic features. The executive profile suggests a possible selective disruption of prefrontal circuits, potentially affecting planning, impulse control, emotional regulation, and sustained attention, which are patterns reminiscent of Huntington disease [42], and also described in schizophrenia [43]. Although subtle early on, these difficulties may become more pronounced with disease progression [44]. Sleep- and circadian-related influences have also been discussed in prior work [45]. Comparable affective–executive patterns have also been described in other rare neurogenetic conditions, including cystic fibrosis and Prader–Willi syndrome [46,47]. These interpretations are descriptive and hypothesis-generating; causal inferences cannot be drawn from our cross-sectional, small-sample, self-report dataset in the absence of normative or control comparisons.

The scoping review indicates that anxiety, depression, apathy, and behavioral dysregulation are recurrently described in individuals carrying WFS1 variants [5,6,7,8,20,21,22,23,24]. Prior literature has explored WFS1-related cellular pathways (e.g., endoplasmic reticulum stress homeostasis and Ca^2+^ signaling) [11,32] and animal work has reported serotonergic alterations [48]; these mechanistic proposals were not assessed in our study and are considered hypothesis-generating rather than explanatory.

Psychological manifestations in WS also resemble patterns reported in other rare and neurogenetic conditions: in cystic fibrosis, pooled adult prevalence estimates of depression and anxiety reach 27.2% and 28.4% [46], and in Prader–Willi syndrome, attentional problems, compulsive behaviors, and mood disorders are common [47]. Case-based descriptions have proposed possible cholinergic contributions in individual patients [28]; such proposals are provisional and beyond the scope of the present dataset. Given heterogeneity in study designs, ascertainment, and the absence of matched controls or normative comparisons in many reports, these psychiatric signals are interpreted descriptively rather than as evidence of differential prevalence versus general adolescent or adult populations. We caution against over-interpreting single case reports and studies without genetic confirmation, control groups, or standardized assessments; signals are not assumed to be equally probative across designs.

Sleep and circadian aspects have been discussed in prior work [12,45], and sleep-disordered breathing has been described in WS [49,50]. We did not measure sleep or circadian variables in this cohort; thus, any sleep- or circadian-related interpretations are hypothesis-generating and not explanatory for our data.

Beyond psychiatric features, fatigue is frequently reported in Wolfram syndrome and plausibly intersects with sleep–wake timing, circadian misalignment, and sleep quality. Although the present dataset was not time-stamped and did not include standardized fatigue scales, future studies should integrate brief fatigue instruments such as the Fatigue Severity Scale, the Functional Assessment of Chronic Illness Therapy–Fatigue, or the Chalder Fatigue Questionnaire, together with chronotype and actigraphy to assess diurnal variation and alignment. Such circadian-informed phenotyping may refine diagnostic interpretation and guide hypothesis-generating chronotherapeutic strategies (e.g., sleep regularity, timed light exposure, melatonin).

### 4.1. Hypotheses and Future Directions

Available findings are consistent with the possibility that the psychiatric manifestations of WS are not incidental but may reflect neurobiological consequences of WFS1-related cellular dysfunction. Taken together, these observations support the notion that WFS1 mutations may increase susceptibility to suicidal and aggressive behaviors, particularly in interaction with environmental and psychological stressors. This perspective aligns with a broader, hypothesis-generating view that executive and emotional dysregulation in WS could arise from a distinct neurobiological substrate. We do not advance mechanistic claims; our dataset did not include molecular or neurophysiological measures, and any biological pathways discussed in prior literature are considered hypothesis-generating only.

Endocrine dysfunction represents another plausible contributor to the psychological burden observed in WS [36]. Hypogonadism, commonly reported in these patients, may result from impaired signaling within the hypothalamic-pituitary-gonadal axis, leading to decreased levels of sex hormones, including testosterone and estrogen [34]. These hormonal deficiencies may negatively affect mood, energy, and motivation, potentially exacerbating depression, apathy, and fatigue. Importantly, such endocrine imbalances could intensify preexisting neurotransmitter dysregulation, further amplifying affective and cognitive symptoms. In line with our results, WS may also include a chronic-fatigue component linked to neuroinflammatory and metabolic dysfunction, expanding its clinical phenotype and reinforcing the need to explore immune-targeted strategies in future studies.

Additionally, neurodegeneration, particularly affecting regions such as the cerebellum (motor coordination) and the prefrontal cortex (emotional regulation and executive functioning), may further compromise emotional stability. This structural deterioration, combined with oxidative stress and neuronal apoptosis, could impair the brain’s capacity for emotional modulation, potentially resulting in irritability, anxiety, and emotional dysregulation [13].

In line with this, experimental work in animal models has shown that preventing mitoNEET-mediated mitochondrial dysfunction significantly attenuates oxidative stress and neuronal apoptosis [39], reinforcing the biological plausibility that mitochondrial pathways contribute to the affective and cognitive manifestations observed in WS.

The interplay among hormonal deficits, neurotransmitter imbalances, and structural brain alterations may contribute to a substantial portion of the psychiatric phenotype in WS. Thus, careful consideration of neuroendocrine interactions is warranted when addressing the psychological manifestations of this syndrome.

DEX observations should be interpreted with caution. Although the pattern suggests relatively preserved social reasoning and behavioral coherence, intermittent executive lapses, particularly under cognitively demanding conditions, should not be underestimated. Even subclinical impairments, when compounded by environmental stressors or disease progression, may significantly impact functional capacity in academic, occupational, or social contexts [8,29]. Such subtle deficits often remain undetected in standard evaluations yet may impose a cumulative burden over time. Given the progressive neurodegenerative course of WS, ongoing monitoring of executive and emotional functions is essential. Longitudinal assessment may facilitate early detection of functional decline and the timely implementation of targeted cognitive or behavioral interventions before more disabling symptoms emerge.

The data obtained through the DEX also inform future directions. Collectively, these findings are compatible with a pattern of executive dysfunction consistent with a frontal-executive syndrome potentially influenced by progressive neurodegeneration in WS [5]. This series of deficits supports, at a descriptive level, the hypothesis that WFS1 dysfunction could compromise higher-order cognitive and emotional regulation via prefrontal network disruption. Importantly, despite notable impairments in impulse control, planning, emotional regulation, and sustained attention, capacities related to social awareness and reality monitoring appeared relatively preserved in most individuals. Participants with WS generally retained the ability to recognize social norms, understand interpersonal consequences, and distinguish real experiences from delusions or hallucinations.

This relative preservation of social judgment and reality orientation contrasts with the cognitive disorganization typically observed in psychotic disorders. There was no consistent evidence of severely disorganized thinking or behavior, nor of profound global cognitive dysfunction. Therefore, the executive profile observed in this cohort is more consistent with a moderate dysexecutive syndrome, suggestive of focal frontal dysfunction rather than a global collapse of reasoning or perceptual coherence.

From a clinical standpoint, the observed executive profile supports exploring targeted interventions. Metacognitive training and cognitive-behavioral approaches may provide compensatory scaffolding for planning and inhibitory deficits [51]. Psychoeducational programs for caregivers may help mitigate family burden [52], and combined cognitive strategies may further enhance outcomes [53]. Routine follow-up should combine self-report (e.g., DEX) with performance-based tests to obtain a comprehensive executive profile [9,10]. A notable gap is the scarce research on caregivers’ mental health in WS and on family-based interventions, despite high dependency in advanced stages; evidence from other rare diseases indicates substantial psychosocial impacts on families [54].

Future research should adopt a biopsychosocial framework that integrates the neurobiological substrates of WS with psychological resilience and family system dynamics. We advocate for longitudinal, multicenter studies integrating neuropsychology with sleep and circadian assessments and biomarkers to clarify how WFS1-related processes may interact with neurotransmitter systems and frontal networks [45,48]. Intervention trials, including metacognitive training, pharmacologic modulation of serotonergic and dopaminergic signaling, and circadian-based strategies, are warranted to test whether targeted therapies can mitigate executive and affective symptoms in WS [51,52,53].

### 4.2. Limitations

This study has several important limitations. First, the scoping review does not provide a quantitative estimate of the prevalence of executive dysfunction across the WS population, and the quality of the included studies was not formally appraised. Second, the DEX cohort is relatively small and drawn from a single national sample, which may limit the generalizability of the findings. Third, the cross-sectional design precludes any causal inferences regarding the relationships among *WFS1* dysfunction, circadian disturbances, and executive outcomes. Fourth, the exclusive reliance on a single self-report instrument introduces potential biases related to subjective perception, recall, and limited insight. Moreover, the absence of objective biological markers—such as molecular, neuroimaging, or electrophysiological data—restricts the ability to validate self-reported outcomes with physiological measures. We did not collect standardized fatigue measures, time-stamped symptom assessments, or chronotype data; this further limits fatigue- and circadian-related inferences and should be addressed in future studies. Finally, the study did not include a formal assessment of circadian rhythms or sleep–wake patterns, which may be relevant to the observed cognitive and behavioral profiles. Future research should aim to incorporate larger, multinational cohorts and employ longitudinal designs integrating neuropsychological testing, circadian and sleep assessments, and biological markers to clarify developmental trajectories and underlying mechanisms.

## 5. Conclusions

The findings of this exploratory study, based on DEX self-report, suggest that individuals with WS often endorse difficulties in attention, inhibitory control, arousal/emotional regulation, and initiation, typically at low-to-moderate frequency. Severe or pervasive executive dysfunction did not predominate; reported problems were generally infrequent and situational (e.g., attentional lapses or fluctuations in arousal). These observations are descriptive signals rather than diagnostic evidence and should be interpreted cautiously given the self-report format, the small cohort, and the absence of normative conversions or matched controls. Nonetheless, they highlight candidate domains of selective executive vulnerability that warrant targeted assessment in future controlled, multimodal, and longitudinal studies.

Important gaps remain in the characterization of the psychological and neurocognitive profile of WS. Prior literature discusses cellular and genetic aspects of WS, but our study did not include molecular or neurophysiological measures; therefore, mechanistic interpretations are hypothesis-generating only. Emerging work has explored potential links between WFS1 and circadian/sleep regulation, which could modulate cognitive and affective functioning; these possibilities were not assessed here and warrant prospective evaluation [12,45,49,50].

Caregiver dynamics and family systems are additional influences that are rarely examined yet may shape resilience or vulnerability in WS [54]. From a clinical standpoint, WS remains a valuable model to study how endocrine, circadian, and neuropsychological factors may converge on executive and emotional regulation, recognizing that current evidence is limited and heterogeneous.

While many participants in this cohort reported a general sense of psychological well-being with selective and mostly infrequent executive difficulties, there is a clear need for larger, longitudinal, and interdisciplinary studies. Future work should define psychological phenotypes with standardized assessments and appropriate comparators, integrate sleep/circadian and neuroendocrine measures, and evaluate targeted interventions adapted to the cognitive and emotional needs of individuals with WS. Such advances may improve clinical care and refine broader models of brain–behavior relationships in rare neurogenetic disorders.

Visual and cognitive deterioration may progress concurrently in WS, with optic neuropathy representing one of the earliest indicators of neurological decline [55].

## Figures and Tables

**Figure 1 diagnostics-15-02338-f001:**
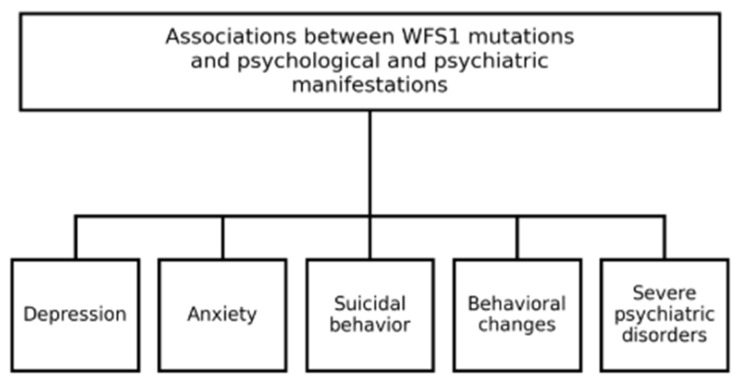
Psychological and psychiatric manifestations associated with WFS1 mutations. Schematic summary of categories identified in the scoping review. Source: author’s elaboration; data synthesized from scope review.

**Figure 2 diagnostics-15-02338-f002:**
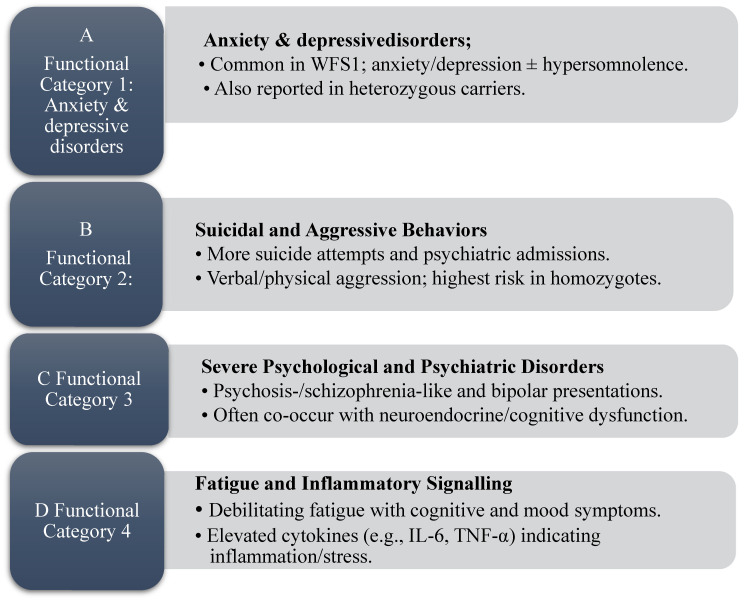
Category summaries derived from the scoping review (panels (**A**–**D**)): (**A**) anxiety and depressive disorders; (**B**) suicidal and aggressive behaviors; (**C**) severe psychological and psychiatric disorders; (**D**) fatigue and inflammatory signaling. Source: Authors’ elaboration.

**Table 1 diagnostics-15-02338-t001:** Key characteristics of the studies included in the scoping review. Columns show study design, sample size, control group, methodology, and a qualitative appraisal of reliability/rigor. This appraisal is descriptive and not a formal quality assessment. Reference numbers [No.] correspond to the main bibliography. Abbreviations: CR, case report; Exp, experimental; Long, longitudinal; Retr (or Retro), retrospective, N/A, not applicable.

No.	Study Design	Sample Size	Control Group	Methodology	Reliability and Rigor Assessment
[5]	Exp.	n = 19	Yes (n = 50)	Multiple standardized cognitive and psychiatric tests	High reliability. Small sample
[7]	Exp.	n = 39	No	Questionnaires and interviews	High reliability and rigor. Medium sample
[8]	Retr.	n = 68	No	Follow-up of previous study	Moderate reliability. No new data collected
[11]	Exp.	N/A	No	Molecular method	High reliability. No human sample
[13]	Retr.	n = 11	No	Clinical data review	Medium reliability. Small sample and no control group
[15]	Retr.	n = 790	No	Cross-sectional study	High reliability. Large sample but no control group
[16]	Retr.	n = 300	No	Correlational analysis	High reliability. Large sample but no control group
[17]	CR	n = 1	No	Clinical case description	Moderate reliability. Single case limits generalizability
[18]	CR	n = 1	No	Clinical case description	Moderate reliability. Single case limits generalizability
[19]	CR	n = 1	No	Clinical case description	Moderate reliability. Single case limits generalizability
[20]	CR	n = 2	No	Clinical case description	Moderate reliability. Small sample and no control group
[21]	Exp.	n = 36	Yes (spouses)	Questionnaires	High reliability. Control group (spouses) provides some comparison
[22]	CR	n = 1	No	WAIS-III and clinical observation	Moderate reliability. Single case limits generalizability
[23]	CR	n = 1	No	Clinical observation	Moderate reliability. Single case limits generalizability
[24]	Exp.	n = 111	Yes (n = 129)	Post-mortem genetic analysis	High reliability. Control group and genetic analysis
[25]	CR	n = 1	No	WAIS and WMS tests	Moderate reliability. Single case limits generalizability
[26]	Exp.	n = 25	No	Questionnaires and interviews	Moderate reliability. Small sample and no control group
[27]	Review	N/A	No	Literature review	Moderate reliability. No original data
[28]	CR	n = 1	No	Clinical case description	Moderate reliability. Single case limits generalizability
[29]	Exp.	N/A	No	iPSC-derived models	Innovative method but no human sample
[30]	Review	N/A	No	Literature review	Comprehensive but no original data
[31]	Exp.	N/A	No	Zebrafish model	Animal model limits human applicability
[32]	Exp.	N/A	No	Genetic and cellular analysis	Moderate reliability. Focus on molecular mechanisms
[33]	Exp./Case	n = 1	No	Genetic and clinical analysis	Moderate reliability. Single case limits generalizability
[34]	Review	n = 86	No	Literature review	Moderate reliability. Comprehensive but no original data.
[35]	Retr.	n = 14	No	Clinical data review	Small sample and no control group
[36]	Long.	n = 39	Yes	Structured interviews and CASI-5	High reliability. Control group and standardized measures.

**Table 2 diagnostics-15-02338-t002:** Functional categories. Note: Sometimes, the same article refers to different issues affecting the people concerned. For this reason, numbers are repeated in some categories. Source: Author’s elaboration.

Category	Summary	References
Severe psychological and psychiatric disorders	Includes the presence of psychiatric conditions such as depression, anxiety, and bipolar disorder. These papers examine potential biological, genetic, and neurological links. Also, addresses cognitive deficits, neurodegeneration, and neuropsychiatric changes in WS, and their impact on daily function and disease progression	[5,6,7,22,24,27,32,36]
Suicidal and aggressive behaviors	WS has been linked on different papers to an increased risk of suicidal and aggressive behaviors	[5,24,27,30,36]
Anxiety and depression disorders	Studies examines anxiety and depressive disorders in individuals with WS and other rare diseases, focusing on the psychological burden of chronic progressive conditions	[13,14]

**Table 3 diagnostics-15-02338-t003:** Sample characteristics of the Wolfram syndrome DEX cohort aged ≥16 years (n = 28). *n* indicates the number of participants, and % (in brackets) their proportion. Age is presented as mean and standard deviation (SD). Education labels reflect the original clinical records; minor overlaps were harmonized to avoid duplication. Participants aged 12–15 years were not included in this table or in normative analyses. Source: Data from the present study.

Characteristic	Value
N (total)	28
Sex: Women, n (%)	13 (46.4)
Sex: Men, n (%)	15 (53.6)
Age, mean (SD)	24.4 (9.7)
IQ band: High, n (%)	13 (46.4)
IQ band: Normal, n (%)	9 (32.1)
IQ band: Low-average, n (%)	4 (14.3)
IQ band: Low/borderline, n (%)	1 (3.6)
IQ band: Intellectual disability, n (%)	1 (3.6)
Education: Higher education, n (%)	9 (32.1)
Education: Basic education, n (%)	9 (32.1)
Education: Middle-level, n (%)	3 (10.7)
Education: Age-appropriate level, n (%)	4 (14.3)
Education: Medium-level, n (%)	2 (7.1)

**Table 4 diagnostics-15-02338-t004:** DEX total score: descriptive statistics (n = 28). Values summarize the Dysexecutive Questionnaire (DEX) total score for the cohort; higher scores indicate greater dysexecutive difficulties. No normative conversions or control-group comparisons were applied. Source: Data from present study.

Measure	DEX Total Score
Mean	30.43
SD	N/A
Median	22.50 (approx.)
IQR (Q1–Q3)	20.00–32.25 (approx.)
Min	≥20.00 (bound)
Max	≤70.00 (bound)

**Table 5 diagnostics-15-02338-t005:** Highest-scoring DEX items in patients with WS: mean. Response scale: 1–5; higher scores indicate greater difficulty. Source: Data from present study.

Question	Executive Dimension	Mean
15	Hyperactivity and restlessness	2.80
18	Difficulties in maintaining attention	2.27
2	Impulsivity (acting without thinking)	2.05
8	Lack of enthusiasm/apathy	2.03
11, 12, 14, 19	Difficulty in showing emotions, anger, repetitive behaviors, indecisiveness	1.99
17,16	Inconsistency between what is said and what is done	1.89, 1.87

**Table 6 diagnostics-15-02338-t006:** Lowest-scoring DEX items in patients with WS: mean. Response scale: 1–5; higher scores indicate greater difficulty. Source: Data from present study.

Question	Executive Dimension	Mean
9	Embarrassing behavior toward others	1.30
3	Confusion with reality (false memories)	1.49
4	Future planning	1.70
6, 7	Confusion between events	1.72

## Data Availability

The datasets generated and analyzed during the current study are available from the corresponding author upon reasonable request. Due to the sensitive nature of the data involving patients with rare diseases, access may be restricted to protect participant confidentiality.

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
