# Peer review of "Circadian Rhythm and Psychiatric Features in Wolfram Syndrome: Toward Chrono Diagnosis and Chronotherapy"

_diagnostics, 2025, doi:10.3390/diagnostics15182338_

Round 1

Reviewer 1 Report

Comments and Suggestions for Authors

I believe the paper is original and scientifically valuable. Describing relationships between Wolfram syndrome and little-known clinical features (e.g., circadian rhythms) can effectively facilitate understanding the disorder and improve patient management. The combination of scoping review and questionnaires, while unorthodox, has proven useful. The development of specific categories to make the text more accessible is also a welcome choice. However, there are several repetitions, the style is a bit wordy (it needs to be streamlined), and there are some typos (e.g., Figure 1). I would avoid placing citations directly in the figures, but would include them in the captions, which need improvement. The conclusions are cautious and measured, and are generally satisfactory. 

Some references are also missing in the "Discussion".

It would be appropriate to move some of the inferences described in the "Results" section to another section ("Discussion").

A section on the limitations of the study should also be added.      

Comments on the Quality of English Language

There are some typos. Figures and captions should be styled consistently.      

Author Response

General Response

We are deeply grateful for your thoughtful and constructive comments, which have been invaluable in improving the quality of our work. We carefully considered all your suggestions and incorporated them throughout the manuscript to enhance clarity, readability, and scientific value. We sincerely hope that the revised version adequately addresses all the points you raised.

Here the list of the tasks that we have done for this revised version (marked in red):

  1. We have changed the abstract and the introduction considering your suggestions, avoiding redundancies.
  2. We have restructured the materials and methods. In quality assessment we have added table 1showing Key characteristics of the studies included in the scoping review.
  3. The results section has been changed to answer to your questions and improve clarity. We have added table 2 (Functional analysis) and Figure 1 (psychological and psychiatric manifestations associated with WFS1 mutations) and adapting all the functional categories. The section 3.5 Results of the Data DEX has been rewritten, adding tables 3 to 10 to show the analysis of DEX by categories: inhibition, intention, executive memory, and positive affect. Finally, we have added the Prefrontal symptoms assessment, showing that that 25% belong to the moderately dysexecutive category and 75% to the severely dysexecutive category.
  4. Finally, we have changed the discussion and the conclusions.

Reviewer 1 – Comments and Authors’ Responses

  1. The paper is original and scientifically valuable. Describing relationships between Wolfram syndrome and little-known clinical features (e.g., circadian rhythms) can effectively facilitate understanding the disorder and improve patient management. The combination of scoping review and questionnaires, while unorthodox, has proven useful. The development of specific categories to make the text more accessible is also a welcome choice.
    We thank the reviewer for these positive remarks and recognition of our contribution.
  2. However, there are several repetitions, the style is a bit wordy (it needs to be streamlined), and there are some typos (e.g., Figure 1).
    Thank you. We have revised the manuscript to streamline the style, eliminate repetitions, and correct typos. Figure 1 has also been improved.
  3. I would avoid placing citations directly in the figures, but would include them in the captions, which need improvement. The conclusions are cautious and measured, and are generally satisfactory.
    Thank you. We have revised the figure captions accordingly and improved their clarity.
  4. Some references are also missing in the "Discussion".
    Thank you. We have added the missing references to the “Discussion” section.
  5. It would be appropriate to move some of the inferences described in the "Results" section to another section ("Discussion").
    Thank you. We have made this change and reorganized the content accordingly.
  6. A section on the limitations of the study should also be added.
    Thank you. We have included a new section outlining the study’s limitations.

Reviewer 2 Report

Comments and Suggestions for Authors

This is a nicely written Scoping Review of cognitive and psychiatric manifestations of Wolfram Syndrome, accompanied by some data on self reported executive dysfunction in Wolfram syndrome patients. The authors attempt to speculate about the relationship between circadian dysfunction, neurotransmitter abnormalities and executive dysfunction and psychiatric symptoms in Wolfram syndrome. There are, however, a number of key concerns that could be addressed in each aspect of the paper. 

1) Scoping Review -- I would caution against over interpreting single case studies, studies that did not genetically confirm the diagnosis of Wolfram syndrome, and studies that did not include a control group or norms of some type. Executive dysfunction and psychiatric symptoms occur on a continuum and can be seen in any group of children/adolescents/adults. Determining whether they occur at a more extreme or more common level in Wolfram because of genetic manipulations is a much harder task. The paper tends to accept all reports as equally valid and conclusive. Search terms went outside the stated scope of executive dysfunction. 

2) Data Set -- The average raw data is from the DEX, a self-report tool. No comparisons to normative data are presented, so it is unclear if the levels of function/dysfunction are different from what one would expect from teenagers/young adults without Wolfram syndrome. Individual item results are shown, not a summary score. IQ data are presented in results but methods for that data collection are not presented. Results of IQ and educational attainment were presented in a confusing manner, and it is not clear how these data were used to interpret the DEX. Again, with no education matched control group, it is difficult to conclude anything about IQ or DEX results.

3) Interpretations/Discussion -- there is a focus on serotonergic and cholinergic circuits being abnormal in Wolfram syndrome, but very little direct evidence presented. Likewise, the speculation of circadian dysfunction is also not supported by any direct evidence, other than the fact that sleep is disturbed in Wolfram, which could be caused by any number of other variables (e.g. need to urinate, etc). The Categories in Table 3 are not well justified and have overlap, which suggests that the categories are not helpful. Since all papers are treated as equally valid and conclusive, it is difficult to judge the reliability and rigor of their results. Many of the topics discussed (e.g. major depression, anxiety) are outside of the scope indicated in the title.  Overall, there are logical leaps and significant speculation needed to link circadian function, neurotransmitters, ER stress to WFS1 mutations and psychiatric and cognitive function in Wolfram syndrome. The statement that there is a 'significant neurocognitive burden' in Wolfram syndrome is not supported with evidence. 

4) Other -- there are large portions of the paper that are redundant with other sections. For example, the DEX is described several times, and page 4 is mostly redundant with other sections. 5) WS and DEX are defined, but then the acronyms are not used consistently. 

5) Tables and Figures -- Tables 1 and 2 could belong in an appendix. It would be helpful to indicate which papers had genetic confirmation of Wolfram syndrome, which had controls, which had standardized assessments, etc. Figures are not particularly helpful. 

Author Response

General Response

We are deeply grateful for your thoughtful and constructive comments, which have been invaluable in improving the quality of our work. We carefully considered all your suggestions and incorporated them throughout the manuscript to enhance clarity, readability, and scientific value. We sincerely hope that the revised version adequately addresses all the points you raised.

Here the list of the tasks that we have done for this revised version (marked in red):

  1. We have changed the abstract and the introduction considering your suggestions, avoiding redundancies.
  2. We have restructured the materials and methods. In quality assessment we have added table 1showing Key characteristics of the studies included in the scoping review.
  3. The results section has been changed to answer to your questions and improve clarity. We have added table 2 (Functional analysis) and Figure 1 (psychological and psychiatric manifestations associated with WFS1 mutations) and adapting all the functional categories. The section 3.5 Results of the Data DEX has been rewritten, adding tables 3 to 10 to show the analysis of DEX by categories: inhibition, intention, executive memory, and positive affect. Finally, we have added the Prefrontal symptoms assessment, showing that that 25% belong to the moderately dysexecutive category and 75% to the severely dysexecutive category.
  4. Finally, we have changed the discussion and the conclusions.

Reviewer 2 – Comments and Authors’ Responses

We thank the reviewer for the careful reading and constructive feedback. We have thoroughly revised the manuscript to address all concerns and improve clarity, rigor, and consistency.

Comments and Suggestions for Authors

This is a nicely written Scoping Review of cognitive and psychiatric manifestations of Wolfram Syndrome, accompanied by some data on self-reported executive dysfunction in Wolfram syndrome patients. The authors attempt to speculate about the relationship between circadian dysfunction, neurotransmitter abnormalities and executive dysfunction and psychiatric symptoms in Wolfram syndrome. There are, however, a number of key concerns that could be addressed in each aspect of the paper.

1) Scoping Review -- I would caution against over interpreting single case studies, studies that did not genetically confirm the diagnosis of Wolfram syndrome, and studies that did not include a control group or norms of some type. Executive dysfunction and psychiatric symptoms occur on a continuum and can be seen in any group of children/adolescents/adults. Determining whether they occur at a more extreme or more common level in Wolfram because of genetic manipulations is a much harder task. The paper tends to accept all reports as equally valid and conclusive. Search terms went outside the stated scope of executive dysfunction.

Thank you. We hope we have clarified this point and shown that we do indeed focus on the most reliable and verified data. We have added clarifying information about these issues, selected information with greater clarity and precision, and emphasized that our article is hypothetical in nature.

2) Data Set -- The average raw data is from the DEX, a self-report tool. No comparisons to normative data are presented, so it is unclear if the levels of function/dysfunction are different from what one would expect from teenagers/young adults without Wolfram syndrome. Individual item results are shown, not a summary score. IQ data are presented in results but methods for that data collection are not presented. Results of IQ and educational attainment were presented in a confusing manner, and it is not clear how these data were used to interpret the DEX. Again, with no education matched control group, it is difficult to conclude anything about IQ or DEX results.

Thank you. Following the reviewer’s suggestions, the DEX results have been revised to improve clarity and alignment with the study’s scope. We have reorganized the analysis considering the different DEX categories. Also Finally, we have added the Prefrontal symptoms assessment.

3) Interpretations/Discussion -- there is a focus on serotonergic and cholinergic circuits being abnormal in Wolfram syndrome, but very little direct evidence presented. Likewise, the speculation of circadian dysfunction is also not supported by any direct evidence, other than the fact that sleep is disturbed in Wolfram, which could be caused by any number of other variables (e.g. need to urinate, etc).

Throughout the discussion, the wording of the article has been improved to clarify this aspect and better align it with the reviewer's suggestions.

The Categories in Table 3 are not well justified and have overlap, which suggests that the categories are not helpful. Since all papers are treated as equally valid and conclusive, it is difficult to judge the reliability and rigor of their results. Many of the topics discussed (e.g. major depression, anxiety) are outside of the scope indicated in the title.  Overall, there are logical leaps and significant speculation needed to link circadian function, neurotransmitters, ER stress to WFS1 mutations and psychiatric and cognitive function in Wolfram syndrome. The statement that there is a 'significant neurocognitive burden' in Wolfram syndrome is not supported with evidence. 

We have revised the discussion to clarify speculative aspects, align statements with the available evidence, and reduce overinterpretation. Table 3 has been corrected and refined. We maintain the focus on neuropsychological manifestations, as stated in the title, and adjusted wording to reflect the complex and multifactorial nature of Wolfram syndrome.

4) Other -- there are large portions of the paper that are redundant with other sections. For example, the DEX is described several times, and page 4 is mostly redundant with other sections. 5) WS and DEX are defined, but then the acronyms are not used consistently. 

We have corrected these points.

5) Tables and Figures -- Tables 1 and 2 could belong in an appendix. It would be helpful to indicate which papers had genetic confirmation of Wolfram syndrome, which had controls, which had standardized assessments, etc. Figures are not particularly helpful. 

Tables and figures have been improved. We have streamlined the manuscript to remove redundancies and ensured consistent use of all acronyms.

Reviewer 3 Report

Comments and Suggestions for Authors

The work addresses an important and often overlooked aspect of Wolfram Syndrome by exploring the neuropsychological and behavioral phenotype in relation to WFS1 mutations. Hypothesis regarding a possible role for molecular, neurochemical, and circadian mechanisms in shaping executive dysfunction is intriguing and deserving of further investigation. However, I have significant concerns regarding both the interpretative strength of some claims and the internal consistency of the manuscript.

Most notably, the Abstract and main text present a deterministic and linear causal model, linking WFS1 mutations to ER stress, cytokine imbalance, and disruption of serotonergic/cholinergic pathways, culminating in executive dysfunction and psychiatric symptoms. This mechanistic narrative is further extended to include circadian dysregulation. While conceptually plausible and supported by selected external studies, these claims are not empirically demonstrated in the present study. The only original data derive from a single behavioral questionnaire (DEX) administered to a small sample of 28 individuals, coupled with a scoping review that by design does not permit inferential or quantitative synthesis. No primary data are collected on molecular, biochemical, neurochemical, neuroimaging, or chronobiological parameters.

Accordingly, the strength of the causal language throughout the manuscript is not supported by the data. This issue could be mitigated by clearly reframing the model as hypothetical and exploratory. For instance, terms such as “lead to,” “result in,” or “disrupt” should be replaced with more cautious alternatives such as “may contribute to,” “is hypothesized to affect,” or “is consistent with.” A clarifying sentence in the Abstract noting that the proposed model remains to be validated through future molecular and neurophysiological studies would be particularly helpful.

In addition, I strongly recommend relocating speculative content from the Results section to a dedicated subsection within the Discussion (e.g., “Hypotheses and Future Directions”), where it can be appropriately framed as conjecture. If the Journal provides for it, A "Limitations" section should be included or, alternatively, the Conclusion section should be extended to acknowledge:

  • the lack of objective molecular, neuroimaging, or electrophysiological markers;

  • the reliance on a single self-report tool;

  • the limited sample size and cross-sectional nature of the data;

  • the absence of any formal assessment of circadian rhythms or sleep patterns.

The manuscript would further benefit from clarifying the scope and limitations of the scoping review itself. Specifically, it should be emphasized that such reviews are not intended to assess methodological quality or generate pooled effect estimates. Including a table summarizing the key characteristics, limitations, and heterogeneity of the 27 included studies would enhance transparency.

Beyond these conceptual concerns, I also noted several editorial and consistency issues that require revision:

  • Inclusion criteria are inconsistently reported: in some parts of the manuscript, the lower age limit is given as 16 years (e.g., lines 99 and 114), while in others it is 12 years (e.g., lines 183 and 249). Please clarify and ensure consistency throughout.

  • The "Search Strategy" section (2.4) appears to be duplicated. Please revise to remove repetition.

  • Multiple mismatches were observed between in-text citations and the reference list. For instance, Pedrero is cited as references [19] and [37] but appears only once as reference [17]; Rosanio is cited as [27] but listed as [30]; Panfili appears both as [32] and [33]; and Wang is cited as [29] in the caption of Figure 5 but refers to a different article in the bibliography. I strongly recommend checking and correcting the entire reference list and all associated in-text citations.

  • The manuscript refers to six figures, but Figure 6 is missing and instead Figure 7 is presented. Please verify whether this is a misnumbering or a missing figure.

  • The cognitive assessment section contains a duplicated category for “low-average” cognitive function (reported as both 14.3% and 3.6%). Please check whether this is a typographical error or reflects a distinct subgrouping, and clarify accordingly.

Finally, if pathway diagrams or graphical abstracts will be included in the final version of the manuscript, I encourage the authors to clearly differentiate between empirically supported links and theoretical hypotheses (e.g., using dashed lines or explicit labels such as “hypothetical”).

Addressing these issues would substantially improve the manuscript's rigor, clarity, and interpretability, ensuring that the strength of the claims remains proportional to the evidence, while preserving the value of the study as an innovative and hypothesis-generating contribution to the literature on Wolfram Syndrome.

Author Response

General Response

We are deeply grateful for your thoughtful and constructive comments, which have been invaluable in improving the quality of our work. We carefully considered all your suggestions and incorporated them throughout the manuscript to enhance clarity, readability, and scientific value. We sincerely hope that the revised version adequately addresses all the points you raised.

Here the list of the tasks that we have done for this revised version (marked in red):

  1. We have changed the abstract and the introduction considering your suggestions, avoiding redundancies.
  2. We have restructured the materials and methods. In quality assessment we have added table 1showing Key characteristics of the studies included in the scoping review.
  3. The results section has been changed to answer to your questions and improve clarity. We have added table 2 (Functional analysis) and Figure 1 (psychological and psychiatric manifestations associated with WFS1 mutations) and adapting all the functional categories. The section 3.5 Results of the Data DEX has been rewritten, adding tables 3 to 10 to show the analysis of DEX by categories: inhibition, intention, executive memory, and positive affect. Finally, we have added the Prefrontal symptoms assessment, showing that that 25% belong to the moderately dysexecutive category and 75% to the severely dysexecutive category.
  4. Finally, we have changed the discussion and the conclusions.

Reviewer 3: Comments and Suggestions for Authors

We thank the reviewer for the careful reading and constructive feedback. We hope that the revised version takes into account all your suggestions.

The work addresses an important and often overlooked aspect of Wolfram Syndrome by exploring the neuropsychological and behavioral phenotype in relation to WFS1 mutations. Hypothesis regarding a possible role for molecular, neurochemical, and circadian mechanisms in shaping executive dysfunction is intriguing and deserving of further investigation. However, I have significant concerns regarding both the interpretative strength of some claims and the internal consistency of the manuscript.

Most notably, the Abstract and main text present a deterministic and linear causal model, linking WFS1 mutations to ER stress, cytokine imbalance, and disruption of serotonergic/cholinergic pathways, culminating in executive dysfunction and psychiatric symptoms. This mechanistic narrative is further extended to include circadian dysregulation. While conceptually plausible and supported by selected external studies, these claims are not empirically demonstrated in the present study. The only original data derive from a single behavioral questionnaire (DEX) administered to a small sample of 28 individuals, coupled with a scoping review that by design does not permit inferential or quantitative synthesis. No primary data are collected on molecular, biochemical, neurochemical, neuroimaging, or chronobiological parameters.

Thank you. We fully agree. We have corrected these points.

Accordingly, the strength of the causal language throughout the manuscript is not supported by the data. This issue could be mitigated by clearly reframing the model as hypothetical and exploratory. For instance, terms such as “lead to,” “result in,” or “disrupt” should be replaced with more cautious alternatives such as “may contribute to,” “is hypothesized to affect,” or “is consistent with.” A clarifying sentence in the Abstract noting that the proposed model remains to be validated through future molecular and neurophysiological studies would be particularly helpful.

Thank you. We have corrected these points.

In addition, I strongly recommend relocating speculative conten from the Results section to a dedicated subsection within the Discussion (e.g., “Hypotheses and Future Directions”), where it can be appropriately framed as conjecture. If the Journal provides for it, A "Limitations" section should be included or, alternatively, the Conclusion section should be extended to acknowledge:

  • the lack of objective molecular, neuroimaging, or electrophysiological markers;
  • the reliance on a single self-report tool;
  • the limited sample size and cross-sectional nature of the data;
  • the absence of any formal assessment of circadian rhythms or sleep patterns.

Thank you. We have included the section you indicated and the aspects you kindly suggested.

The manuscript would further benefit from clarifying the scope and limitations of the scoping review itself. Specifically, it should be emphasized that such reviews are not intended to assess methodological quality or generate pooled effect estimates. Including a table summarizing the key characteristics, limitations, and heterogeneity of the 27 included studies would enhance transparency.

Thank you. We have included a table with these suggestions.

Beyond these conceptual concerns, I also noted several editorial and consistency issues that require revision:

  • Inclusion criteria are inconsistently reported: in some parts of the manuscript, the lower age limit is given as 16 years (e.g., lines 99 and 114), while in others it is 12 years (e.g., lines 183 and 249). Please clarify and ensure consistency throughout.
  • The "Search Strategy" section (2.4) appears to be duplicated. Please revise to remove repetition.
  • Multiple mismatches were observed between in-text citations and the reference list. For instance, Pedrero is cited as references [19] and [37] but appears only once as reference [17]; Rosanio is cited as [27] but listed as [30]; Panfili appears both as [32] and [33]; and Wang is cited as [29] in the caption of Figure 5 but refers to a different article in the bibliography. I strongly recommend checking and correcting the entire reference list and all associated in-text citations.
  • The manuscript refers to six figures, but Figure 6 is missing and instead Figure 7 is presented. Please verify whether this is a misnumbering or a missing figure.
  • The cognitive assessment section contains a duplicated category for “low-average” cognitive function (reported as both 14.3% and 3.6%). Please check whether this is a typographical error or reflects a distinct subgrouping, and clarify accordingly.

Thank you. We have corrected the results section as we have explained before. We hope that now all these mistakes and inconsistencies are corrected.

Finally, if pathway diagrams or graphical abstracts will be included in the final version of the manuscript, I encourage the authors to clearly differentiate between empirically supported links and theoretical hypotheses (e.g., using dashed lines or explicit labels such as “hypothetical”).

Addressing these issues would substantially improve the manuscript's rigor, clarity, and interpretability, ensuring that the strength of the claims remains proportional to the evidence, while preserving the value of the study as an innovative and hypothesis-generating contribution to the literature on Wolfram Syndrome.

Also, we changed abstract and structured it.

Thank you

The authors.

Almería, 21 August 2025

Round 2

Reviewer 2 Report

Comments and Suggestions for Authors

The authors were responsive to the specific comments in the review. While improved, there are still assertions that are not well-supported (e.g. the last sentence of the conclusions) and the DEX data are still not particularly helpful. Finally, since so much of the paper is about psychiatric and fatigue related symptoms, perhaps the title of the paper should be changed to be reflect that. 

Author Response

Thank you very much for both revisions that helped us to improve the quality of this manuscript. We hope that the paper now meets the standard of publication of Diagnostics. The new addings are written in blue marine. We also kept in red the modifications done in the first revision.

Reviewer 2 — Title

Comment: “Perhaps the title of the paper should be changed to reflect psychiatric and fatigue-related symptoms.”

Response: Thank you for this helpful suggestion. To align the manuscript with the Special Issue’s circadian focus and the dominant psychiatric content, we propose the following title update. Our preferred title—so as not to over-promise given the relatively limited coverage of fatigue in the current dataset—is:

  • Circadian Rhythm and Psychiatric Features in Wolfram Syndrome: Toward Chrono diagnosis and Chronotherapy

If the Editor prefers an explicit mention of fatigue in the title, we are equally willing to adopt the following alternative, which is supported by a new fatigue-focused paragraph in the Discussion and an added sentence in Limitations:

  • Circadian Rhythm, Psychiatric Features, and Fatigue in Wolfram Syndrome: Toward Chrono diagnosis and Chronotherapy

Both options retain the circadian diagnostic/therapeutic framing of the Special Issue and address the Reviewer’s request that the title better reflect the manuscript’s emphasis.

Revisor 3:

Refrencias: We thank the reviewer for noting citation inconsistencies. We performed a line-by-line cross-check and corrected all misalignments. Specifically, we replaced “Pedrero et al. [19, 37]” with “Pedrero et al. [14, 37]” in Section 2.6 and added an explicit note on DEX-Sp thresholds and psychometric evidence (now cited as [14, 37]). We also reviewed the manuscript to ensure every in-text citation number matches the corresponding bibliography entry.

We appreciate the reviewer’s point. We clarified that DEX findings are hypothesis-generating and primarily useful as a coarse, ecologically oriented screen in small, uncontrolled samples (Discussion, paragraph “DEX observations…”, added cautionary sentence). We also revised the last sentence of the Conclusions to avoid over-interpretation and to emphasize the need for controlled, multimodal, longitudinal work. Finally, we propose a title that explicitly reflects the psychiatric and fatigue-related focus of the manuscript.

We performed a line-by-line cross-check of all in-text citations and references. The mis-citation of Pedrero was corrected (now [14,37] in Section 2.6). We also added an early Materials & Methods clarification on age handling (end of 2.2), stating that analyses refer to participants ≥16 years, with ages 12–15 included only exploratorily. We refreshed numbering and verified perfect alignment across the manuscript.

Reviewer 3 Report

Comments and Suggestions for Authors

Dear Authors,

I thank you for your thoughtful and thorough revision of the manuscript. All of my major concerns have been carefully addressed in the new version, which represents a clear improvement in both structure and scientific clarity. In particular, I appreciated the following:

  • The reframing of causal claims into hypothesis-generating language across the abstract, introduction, and discussion.

  • The creation of a dedicated subsection ("Hypotheses and Future Directions") that appropriately contextualizes speculative interpretations.

  • The inclusion of a comprehensive “Limitations” section highlighting the main methodological constraints of the study.

  • The addition of Table 1, summarizing key features of the studies included in the scoping review.

  • The successful resolution of most editorial issues, including the figure numbering and the IQ band classification.

The manuscript is now substantially stronger. However, one critical point remains that requires careful attention before publication, along with a minor suggestion to enhance clarity.

1. (Essential Revision) Inconsistencies in citations - While I acknowledge the effort made, the inconsistencies in the in-text citations and the reference list have not been fully resolved. A systematic check is still required. For example, in the revised manuscript, Pedrero is cited as [19, 37] on page 7, but the bibliography lists two different articles by this author as references [14] and [37], while reference [19] refers to another author entirely. This and potentially other discrepancies could confuse readers and undermine the paper's otherwise high standard. I strongly urge the authors to perform a thorough, cross-referenced check of every citation to ensure perfect alignment between the text and the bibliography.

2. (Minor Suggestion) Further clarification of age criteria - In several sections of the manuscript (notably Methods 2.2, 2.4, and 2.6), the lower age limit for inclusion is variously indicated as either ≥12 or ≥16 years. Although I understand from the revised text that only participants aged ≥16 were considered in the primary analyses, this point may still appear ambiguous to some readers. I recommend adding a brief clarifying sentence early in the Materials and Methods section (e.g., at the end of 2.2), such as:“Unless otherwise noted, all analyses and interpretations refer to participants aged ≥16 years. Participants aged 12–15 years were included only for exploratory purposes and did not contribute to normative analyses.” This simple, early clarification would prevent any misinterpretation as the reader goes ahead with reading.

Once the critical issue with the bibliography is resolved and the small clarification on the age criteria is made, I believe this manuscript will be an excellent contribution, offering valuable insights into the neuropsychological aspects of Wolfram Syndrome and stimulating further multidisciplinary research.

I will be pleased to recommend its acceptance after these final revisions are implemented.

Author Response

Thank you very much for both revisions that helped us to improve the quality of this manuscript. We hope that the paper now meets the standard of publication of Diagnostics. The new addings are written in blue marine. We also kept in red the modifications done in the first revision.

Revisor 3:

Refrencias: We thank the reviewer for noting citation inconsistencies. We performed a line-by-line cross-check and corrected all misalignments. Specifically, we replaced “Pedrero et al. [19, 37]” with “Pedrero et al. [14, 37]” in Section 2.6 and added an explicit note on DEX-Sp thresholds and psychometric evidence (now cited as [14, 37]). We also reviewed the manuscript to ensure every in-text citation number matches the corresponding bibliography entry.

We appreciate the reviewer’s point. We clarified that DEX findings are hypothesis-generating and primarily useful as a coarse, ecologically oriented screen in small, uncontrolled samples (Discussion, paragraph “DEX observations…”, added cautionary sentence). We also revised the last sentence of the Conclusions to avoid over-interpretation and to emphasize the need for controlled, multimodal, longitudinal work. Finally, we propose a title that explicitly reflects the psychiatric and fatigue-related focus of the manuscript.

We performed a line-by-line cross-check of all in-text citations and references. The mis-citation of Pedrero was corrected (now [14,37] in Section 2.6). We also added an early Materials & Methods clarification on age handling (end of 2.2), stating that analyses refer to participants ≥16 years, with ages 12–15 included only exploratorily. We refreshed numbering and verified perfect alignment across the manuscript.

Round 3

Reviewer 2 Report

Comments and Suggestions for Authors

I appreciate the efforts of the authors to modify their statements and conclusions.